# Kaolinite-Magnesite Based Ceramics. Part I: Surface Charge and Rheological Properties Optimization of the Suspensions for the Processing of Cordierite-Mullite Tapes

**Aghiles Hammas [1]** [ID]**, Gisèle Lecomte-Nana [2],*, Nadjet Azril [3], Imane Daou [2], Claire Peyratout [2] and Fatima Zibouche [1]**

[1] Laboratoire de Traitement et Mise en Forme des Polymères, Faculté des Sciences, Université M'Hamed Bougara Boumerdes, Boumerdes 35000, Algerie; aghiles.hammas@univ-boumerdes.dz (A.H.); zibouchefatima@yahoo.fr (F.Z.)

[2] Institut de Recherche sur les Céramiques (IRCER, UMRCNRS 7315), ENSIL-ENSCI, Université de Limoges, CEC, 12 rue Atlantis, 87068 Limoges, France; imane.daou@unilim.fr (I.D.); claire.peyratout@unilim.fr (C.P.)

[3] Laboratoire Génie Physique des Hydrocarbures–Simulation et rhéologie des Fluides Complexes, Université M'Hamed Bougara Boumerdes, Boumerdes 35000, Algerie; nadjet.n@gmail.com

* Correspondence: gisele.lecomte@unilim.fr

**Abstract:** The present study aimed at investigating the influence of the concentration of sodium silicate and sodium hexametaphosphate on the dispersion of an aqueous kaolinitic clay slurry regarding further use for the tape casting process. The zeta potential of the kaolinitic clay slurry matched the requirements for tape casting. The addition of magnesite in the kaolinitic slurries tended to increase the zeta potential towards the required limit values. Despite this, the further addition of surfactants allowed improving the zeta potential in agreement with the tape casting conditions. Accordingly, the rheological behavior, under continuous and oscillatory flow conditions, of various mixtures of magnesite and a kaolinitic clay was studied. Regarding the pH and the zeta potential measurements, the E–F attraction prevailed at low pH value, and F–F or E–E attraction was predominant at high pH value. All slurries exhibited a shear thinning behavior, which was well-correlated by the Herschel–Bulkley model. It appeared that the best stability for the kaolinitic clay slurries was obtained while using 0.4 mass% and 1.2 mass% of sodium hexametaphosphate and sodium silicate, respectively. An increase in the magnesite concentration above 6 mass% led to a complex behavior with low cohesion energy due to the occurrence of soluble complexes.

**Keywords:** kaolinitic clay; magnesite; slurries; Na silicate; Na hexametaphosphate; rheology

## 1. Introduction

In general, ceramic slurries contain inorganic components (clay minerals, quartz, feldspar, and oxide), water or organic solvents, and most frequently, organic or inorganic additives (dispersants, plasticizers, fluidizers, antifoams, lubricants, etc.). The stabilization of slurries is a critical point in the ceramics industry [1,2] regarding the target shaping process, microstructure, and properties of use. Different electrolytes, polymers, and surfactants had been introduced within slurries in order to obtain stable and homogenous colloidal systems [3]. Indeed, some papers addressed the specific case of the colloid chemistry of clay minerals (i.e., bentonite, talc, palygorskite, and halloysite) with industrial issues. Among all the clays used, kaolins and kaolinitic clays are the most important raw materials for silicate ceramics, due to their abundance and low costs.

Kaolin is a commercial term used to describe a white clay mainly composed of kaolinite ($Si_2O_5Al_2(OH)_4$) with a percentage higher than 80 mass%. For clays containing kaolinite between 50 and 80 mass%, they are considered as kaolinitic clays. In clays, the clay minerals are usually associated with other minerals (accessory minerals), such as quartz, feldspar, and mica (sericite, muscovite, etc.) [4]. Kaolinite mineral has a 1:1 or TO layer type with one octahedral (O) alumina layer ($AlO_6$) and one tetrahedral (T) silica layer ($SiO_4$). Furthermore, The edges represented by Al–OH (aluminol) for the octahedral layer and by Si–OH (silanol) for the tetrahedral layer; these sites are more or less protonated through the hydroxyl groups at the surface, depending on the pH value of the medium [5–7]. Gupta et al. [8] concluded that the silica-like face is negatively charged at pH > 4, while the alumina-like face is positively charged at pH < 6, and negatively charged at pH > 8. It is known that clay-based aqueous suspensions lead to a particular structure, for example, single layers, discrete particles, or aggregates. Considering the surface charge, there are three modes of particle aggregation: edge–to-face (E–F), face–to-face (F–F), and edge–to-edge (E–E). From the literature, it had been concluded that this aggregation and surface charges were widely dependent on the pH value of the slurries [9]. As reported previously, when a negatively charged dispersant is added to aqueous clay system, it adsorbs on the external particle surface and makes the surface charge more negative, inducing electrostatic or steric repulsion between particles [1,10]. Among the dispersants that have been introduced to disperse clay particles by electrostatic interactions, sodium silicate ($Na_2SiO_3$) and sodium hexametaphosphate ($(NaPO_3)_6$) appeared to provide an efficient dispersion effect [1,7,11–17]. Both dispersants adsorbed onto the surface charge of particles by the Van Der Waals interactions and hydrogen bondings, which induce repulsion/attraction interactions [14,18,19]. On the edges, these anions can adsorb to the central cations (Si and/or Al) and increase the coordination number hereby. As multivalent anions, free negatively charged sites shift zeta potential to more negative values. The performance of dispersant addition is determined by dispersion homogeneity, stability, and rheology [20].

Rheological characterization is the study of the deformation of materials undergoing a controlled tunable force, which is carried out using numerous simple controlled methods: steady shear, stress relaxation, creep, oscillatory shear, and steady extension. The rheological behavior of a system is influenced by several parameters, including solid loading, particle morphology, particle size distribution, particle interactions (edge–edge, face–face or edge–face interactions), and chemistry of the dispersions [21]. Steady-state rheology measurements give the relationship between shear stress and shear strain or shear rate. The internal stresses are directly related to the applied force over the studied slurry regardless of their origin; they produce deformation between intermolecular structures. Meanwhile, the strain is defined as the relative deformation, i.e., the deformation per unit length. The length used is the effective size over which the deformation occurs. In this study, all flow curves of slurries are described by the Herschel–Bulkley ($\tau = \tau_0 + K \times \dot{\gamma}^n$) model, which are defined by rheological parameters, for example, yield stress, consistency index, and power-law index.

Furthermore, Bitterlich et al. [22] had suggested that the viscoelastic behavior can complete the flow-curve study by imposing small-amplitude oscillatory shear stress, which provided additional information about clay aggregation. The two parameters that characterize the viscoelastic behavior of dispersions are: (i) the elastic or storage module $G'$, which describes elastic properties of the slurry; (ii) the viscous or loss module $G''$, which gives the applied stress by producing viscous deformations.

The most recent study suggests that magnesium oxide had a greater influence on the sodium hexamethaphosphate-kaolin interactions, which have an anti-dispersant performance in the dispersions [23]. Therefore, the MgO is usually obtained by magnesite decomposition at high temperature [24]. In the present work, the effect of magnesite to the aqueous kaolinitic clay dispersion may cause a decrease or increase in the repulsive force interactions between the solid particles in the slurry, which directly affect the rheological parameters of the medium, as well as its pH. Magnesite is the mineral name for magnesium carbonate ($MgCO_3$), it is described as a light, white, amorphous, and odorless powder capable of odor absorption. It is considered as the original source for magnesium oxide by thermal decomposition in the range 230–680 °C [25]. Moreover, suspensions used for the

subsequent shaping process will contain a plasticizer and binder. All viscosity values of suspensions of the KT2/MgCO$_3$ mixture at a specific shear-rate (10 s$^{-1}$) have to range between 0.3 and 3 Pa·s, which is required for the tape casting process.

In the present study, continuous shear was applied to concentrated aqueous kaolinitic clay dispersions (50 mass%), which were dispersed with sodium silicate and sodium hexametaphosphate (0.4 to 2 mass%). In addition, the effects of MgCO$_3$ addition (3, 6, and 12 mass%) on the rheological properties of the as-obtained slurries were evaluated by continuous and oscillatory shear. Moreover, the pH measurements were carefully controlled in order to examine the influence of the surface charge on the rheological properties of slurries. The optimized slurries were finally tested for the tape casting process.

## 2. Materials and Methods

### 2.1. Raw Materials

In this study, a kaolinitic clay (KT2) and magnesite (MgCO$_3$) were used as raw materials. The clay noted KT2 was obtained from the Tamazert site (El Milia, Jijel, Algeria) and supplied by the Algerian company, Soalka. The magnesite was supplied by CFM Minerals (Castellon, Spain). S.A. Spain. The raw materials chemical compositions, BET specific surface area, and density are shown in Table 1.

**Table 1.** Chemical composition (XRF analyses) and other characteristic physical properties of the starting powders.

| Raw Material | SSA (BET) (±0.1 m$^2$·g$^{-1}$) | Density (g·cm$^{-3}$) | Chemical Composition (mass%) | | | | | | | | |
|---|---|---|---|---|---|---|---|---|---|---|---|
| | | | SiO$_2$ | Al$_2$O$_3$ | MgO | Fe$_2$O$_3$ | TiO$_2$ | CaO | Na$_2$O | K$_2$O | Loss on Ignition at 1050 °C |
| KT2 | 25.0 | 2.6 | 47.8 | 32.9 | 0.6 | 3.5 | 0.5 | 0.1 | 0.1 | 2.9 | 11.5 |
| Magnesite | 11.7 | 2.9 | 3.2 | 0.1 | 42.7 | 0.4 | - | 3.2 | - | - | 50.4 |

Sodium hexametaphosphate (65–70% P$_2$O$_5$ basic, 71,600 Aldrich) and sodium silicate (Biochem Chemopharma, product code: 319160500), noted as NaHMP and NaSi, respectively, were used as powder dispersants in distilled water. Organics additives, such as binder and plasticizer, were introduced to the optimized suspensions of KT2-magnesite mixtures, polyvinyl alcohol (PVA 2200, VWR Prolabo, Geldenaaksebaan, Belgium), and polyethylene glycol (PEG 300, VWR, Prolabo, Germany), respectively.

### 2.2. Characterization Techniques

The particle size distribution was determined using Laser Scattering equipment, Horiba LA-950 V2 (Kyoto, Japan). In this study, before each measurement, 0.5 g of KT2 powder was added to 10 mL of NaHMP solution (1 g/L), and the same quantity of magnesite was added to 10 mL of ethanol. Moreover, both suspensions were de-agglomerated under ultrasound (37 kHz) for 1 min. All the measurements were carried out at room temperature.

The specific surface area was determined by the nitrogen adsorption (BET method) using Quantachrome NovaWin Instruments (Quantachrome Instruments, Boynton Beach, FL, USA). Before testing, the samples were dried for 24 h at 110 °C and then degassed at 250 °C for 3 h. The solid density of clay and magnesite powders was estimated using helium pycnometry (Micromeritics AccuPyc II 1340 device from Micromeritics, Norcross, GA, USA).

The phase purity of the samples was examined by X-ray diffraction (XRD) on a Bruker D8-advance X-ray diffractometer (Bruker, Karlsruhe, Germany) Cu K$\alpha_1$ radiation (1.5418 Å). The operating voltage and current were maintained at 40 KV and 40 mA, respectively. The XRD analysis was performed in the range 2° to 60° (angle 2 theta), with the step of 0.02° and a scanning time of 1.79 s per step.



Thermal analysis is generally used to investigate both qualitative and quantitative properties of materials under heat treatment and controlled atmosphere. Thermogravimetric analysis (TG) and differential thermal analysis (DTA) were measured using the TGA/DSC 3+ STAR System (METTLER TOLEDO, Viroflay, France) at a heating rate of 5 °C/min from room temperature to 1400 °C (1000 °C for magnesite) under a dry air atmosphere (for all experiments Pt crucibles were used).

## 2.3. Dispersions Preparation

The aqueous KT2/$MgCO_3$ suspensions (50 mass%) were prepared according to the following steps: (i) mixing distilled water and dispersants, (ii) adding magnesite at different concentrations (3, 6, and 12 mass%), and finally (iii) adding different KT2 amounts. In order to achieve complete homogenization of suspension components, magnetic stirring was performed for 6 h at 800 rpm. After mixing, pH values were measured by introducing a pH electrode (HI 2211 pH/ORP meter, Hanna Instruments Ltd., Leighton Buzzard, UK) in dispersions, the error associated with the pH measurement was ±0.01. Before casting, the additives, such as binder and plasticizer, were introduced to the suspension mixture, polyvinyl alcohol (PVA 2200) and polyethylene glycol (PEG 300), respectively, which was ground in a planetary mill during 16 h at 100 rpm. In this study, the ratio binder/plasticizer being equal to 1 by weight. A solution of 0.16 g·mL$^{-1}$ binder was prepared by incorporating PVA slowly in deionized water, which was previously heated at 80 °C. During the addition of PVA, the solution was continuously stirred in order to avoid the formation of a gel. Prior to the tape casting, the slurries were de-aired on rollers during 22 h and sieved at 100 μm to ensure the elimination of the non-dissolved binder or impurities.

## 2.4. Acoustophoresis Analysis

The zeta potential analysis of the KT2/$MgCO_3$ mixture particles was performed using an Acoustosizer IIS from Colloidal Dynamics (Ponte Vedra Beach, FL, USA). These measurements need a minimum volume of 150 mL and 5 mass% of concentration. In addition, all slurries were stirred in order to prevent sedimentation. Hydrochloride acid (HCl) and sodium hydroxide (NaOH) solutions of 0.1 M were used to shift the pH value of the studied suspensions over the pH range from 2 to 12.

## 2.5. Rheological Measurements

The rheological experiments were carried out at 25 °C using an Anton Paar Physica MCR301 Rheometer (Anton Paar, Ostfildern, Germany) with controlled coaxial cylinder geometry (radius internal = 13.3250 mm, radius external = 14.4645 mm, cone angle = 120°, and sample volume 19.0 cm$^3$). Flow curves were obtained under continuous flow conditions by increasing the shear rate from 0.001 to 1000 s$^{-1}$ for 150 s.

The viscoelastic properties were studied under oscillatory or dynamic shear, for aqueous slurries containing optimized amounts of dispersants (1.2% NaSi or 0.4% NaHMP) with and without magnesite (0, 3, 6, and 12 mass%). For stress sweeps, the elastic and viscous moduli were determined as a function of the applied stress (0.01 to 10 Pa) at an oscillation frequency of 10 s$^{-1}$. Once an oscillation stress value was equal to 0.1 Pa (linear viscoelasticity region), a frequency sweep was performed from 0.1 to 100 s$^{-1}$. Moreover, the cohesive energy ($Ec$) for suspensions mixture dispersed with an optimal amount of dispersants was calculated according to Equation (1).

$$Ec = \frac{1}{2} G'_c \gamma_c^2 \tag{1}$$

where $G'_C$ is the critical elastic modulus, and $\gamma_c$ is the critical strain. Both parameter values were extracted from oscillatory strain sweep data, which determined by stress sweep results.

## 3. Results and Discussion

### 3.1. Raw Materials Characterizations

The XRD patterns obtained for samples KT2 and $MgCO_3$ are presented in Figure 1. The results indicate that the KT2 sample contains mainly kaolinite (JCPDS: 04-016-1460) and secondary phases as muscovite (JCPDS: 01-082-3725) and quartz (JCPDS: 04-015-8431). The $MgCO_3$ sample consists of magnesite (JCPDS: 04-014-4830), with a small amount of calcite (JCPDS: 04-020-5890).

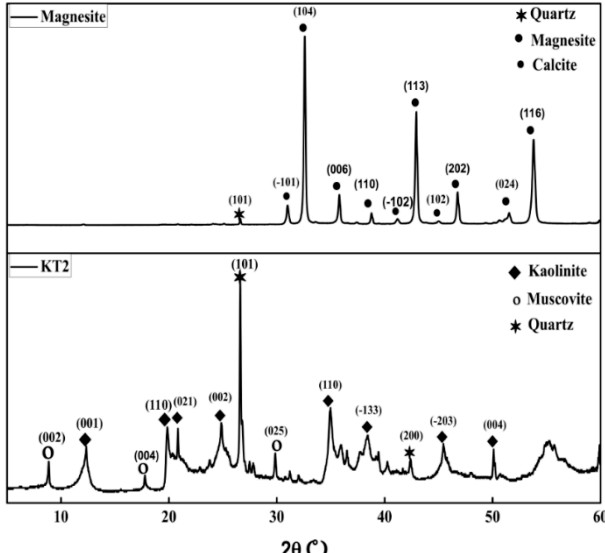

**Figure 1.** XRD patterns of raw materials.

The DTA and TG curves of KT2 (Figure 2a) indicate several phenomena:

(i) A weak endothermic peak below 200 °C associated with a mass loss of 1.4% due to the evaporation of physically adsorbed water (dehydration);

(ii) A large endothermic peak centered between 400 to 550 °C, which is related to the dehydroxylation of kaolinite (see Equation (2)) [26] giving rise to a 9.6 mass% loss;

(iii) An exothermic peak in the temperature range 940–1000 °C, which is mainly due to the structural reorganization of metakaolinite (mullite formation, Equation (3)); the latter phenomenon does not induce any mass variation, but there is the occurrence of an endothermic shoulder or hump prior to the mullite crystallization reaction. This trend is generally observed when the starting kaolinitic lay contains iron and/or muscovite as secondary phases [27,28].

$$Al_2(OH)_4Si_2O_5 \text{ (kaolinite)} \rightarrow Al_2Si_2O_7 \text{ (metakaolinite)} + 2H_2O \tag{2}$$

$$2(Al_2Si_2O_7) \text{ (metakaolinite)} \rightarrow 2Al_2O_3 \cdot 3SiO_2 \text{ (mullite)} + SiO_2 \tag{3}$$

On the other hand, the DTA and TG curves of magnesite (Figure 2b) indicate two phenomena:

(i) A large endothermic peak at the temperature range of 560 to 600 °C associated with a mass loss of 37.5% due to the decomposition of $MgCO_3$ to the MgO and $CO_2$ (see the decarbonation Equation (4)), in addition, an endothermic shoulder occurred in this temperature range, which may result from the loss of small molecules ($Mg(OH)_2$ based);

(ii) A weak endothermic peak at 710 °C was mainly due to the decomposition of calcium carbonate (calcite: $CaCO_3$) into the CaO and $CO_2$ Equation (5) [29], which gives rise to a 4 mass% loss. The presence of $CaCO_3$ on the magnesite sample was confirmed by XRD analysis.

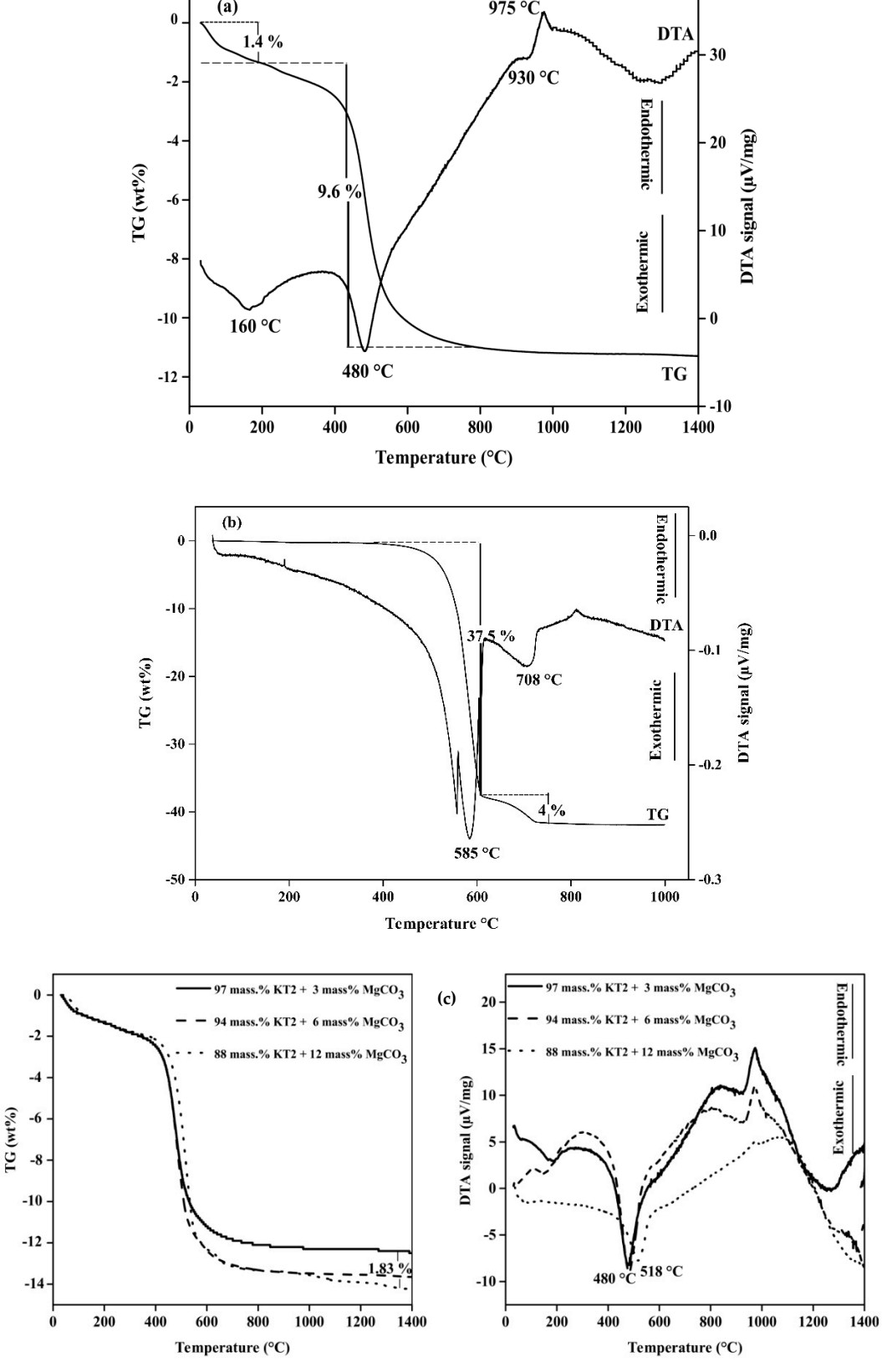

**Figure 2.** Differential thermal analysis (DTA) curves of (**a**) KT2, (**b**) MgCO$_3$, and (**c**) KT2-MgCO$_3$ mixture.

The influence of MgCO$_3$ added (3, 6, and 12 mass%) on DTA and TG curves of kaolinitic clay are also investigated (Figure 2c). The curves indicate that increasing of adding MgCO$_3$ form 6 to 12 mass%

shifted the maximum of the peak position from 480 to 518 °C due to the residual bonds between the layers leads to the unequal location of OH groups [24].

$$MgCO_3 \text{ (s)} \rightarrow MgO \text{ (s)} + CO_2 \text{ (g)} \tag{4}$$

$$CaCO_3 \text{ (s)} \rightarrow CaO \text{ (s)} + CO_2 \text{ (g)} \tag{5}$$

The specific surface area (SSA) of both samples is reported in Table 1. SSA of KT2 was found to be $25.0 \pm 0.1 \text{ m}^2 \cdot \text{g}^{-1}$, in agreement with other reports, for example, [30]. However, the BET surface area of magnesite powder is estimated to be $11.7 \pm 0.1 \text{ m}^2 \cdot \text{g}^{-1}$.

Many previous researchers have indicated that the particle size is an important factor in rheology behavior [31,32]. The results of the granulometry analysis (Figure 3c) indicate that $d_{50}$ of KT2 particles is equal to 14.22 µm, with $d_{10}$ and $d_{90}$ equal to 3.02 and 42.73 µm, respectively. According to Merkus, H [33], these are considered as medium particles. The average particle size ($d_{50}$) of magnesite was 1.02 µm, with $d_{10}$ and $d_{90}$ equal to 0.70 µm and 1.57 µm. According to this result, magnesite sample have fine particles. The latter trend is correlated with SEM observations of KT2 and magnesite, as shown in Figure 3a,b. Note that the platelets and tubular-like shape of the kaolinitic clay particles involve the stacking of several clay mineral sheets. This fact could justify the higher SSA obtained with KT2 compared to the SSA value of magnesite.

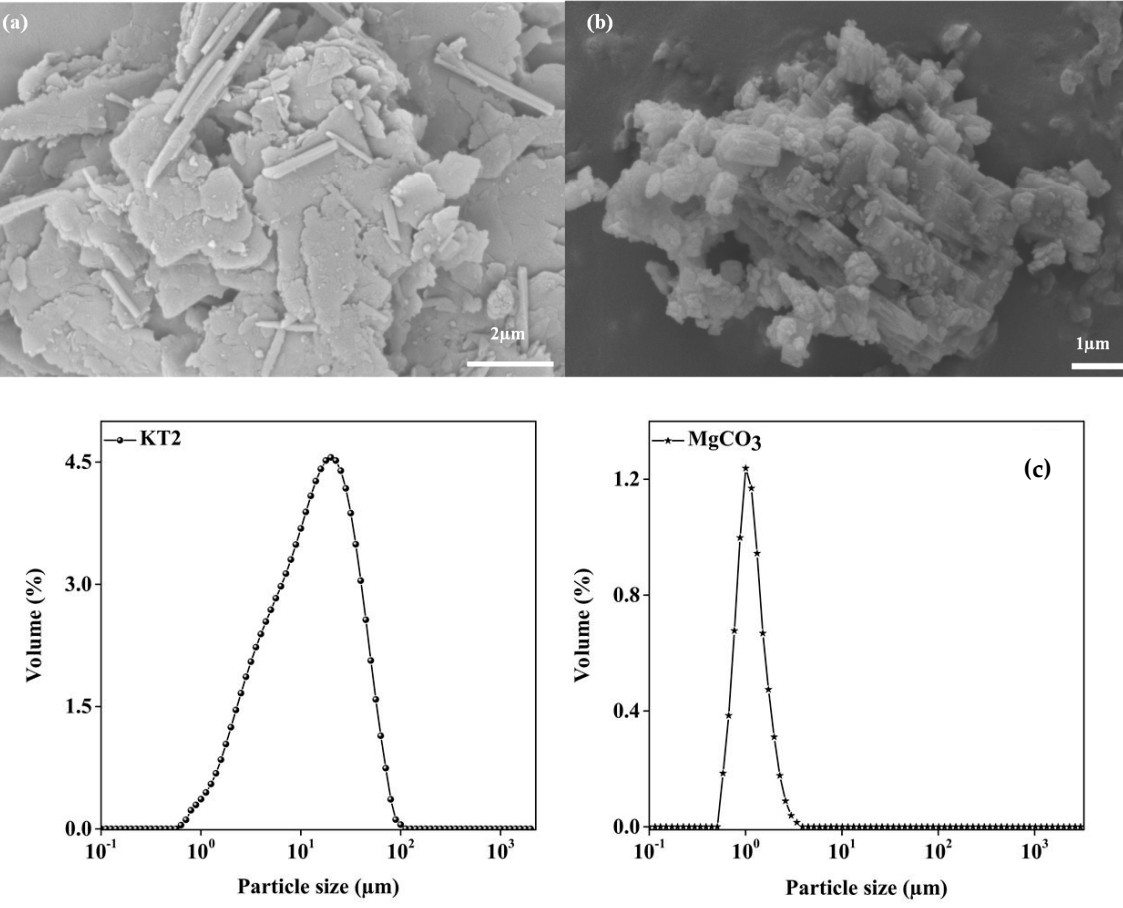

**Figure 3.** SEM micrographies (**a**) KT2 and (**b**) magnesite, and particle size distribution (**c**) of the raw materials.

### 3.2. Zeta Potential Measurements

Usually, the stability and homogeneity of slurries mainly depend on the zeta potential value of particles, since it is correlated with particle surface charge. Then, the flocculation or dispersion

phenomenon within slurries may vary with zeta potential evolution, as indicated in previous studies [34,35]. This behavior attributed to the adsorption of H$^+$ and OH$^-$ ions onto the surface of the particles. However, it can be seen from the data presented in Figure 4a that the zeta potential values of KT2 slurries, regardless of the MgCO$_3$ content, became more negative with increasing pH values from 2 to 12.

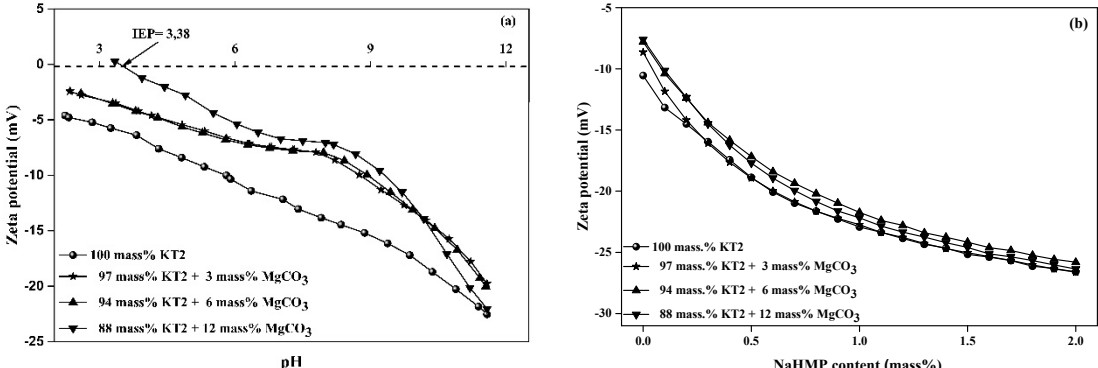

**Figure 4.** Zeta potential-pH behavior of KT2/MgCO$_3$ mixture slurries (**a**) as a function of pH and (**b**) with dispersant NaHMP.

The zeta potential of KT2 slurry varied from −5 to −23 mV with increasing pH values from 2 to 12. For the slurry with 12 mass% of MgCO$_3$, the zeta potential varied from 0.28 to −22 mV in the same range of pH. Consequently, it is noted that the zeta potential became less negative with increasing concentration of the magnesite at acidic pH. This can be explained by the adsorption of positive charges on the KT2 particles due to the dissolution of MgCO$_3$, for example, Mg$^{2+}$ and Mg(OH)$^+$, which involves high aggregation. These observations appear to be in good agreement with previous works that indicate the dissolution mechanism of magnesite in aqueous solution. The Mg$^{2+}$ was the main species in the pH < 9 solution, its concentration decrease with increasing pH above 9, while the concentration of Mg(OH)$^+$ and Mg(OH)$_2$ increased with increasing solution pH. In pH > 10.5 solutions, most of the dissolved carbonate was converted to Mg(OH)$_2$ (Equations (6)–(8)) [36,37].

$$MgCO_3 \; (aq.) + H^+ \rightarrow Mg^{2+} + HCO_3^- \tag{6}$$

$$Mg^{2+} + OH^- \rightarrow MgOH^+ \tag{7}$$

$$MgOH^+ + OH^- \rightarrow Mg(OH)_2 \tag{8}$$

It is also noted that the isoelectric point (IEP) of KT2 slurry containing 12 mass% of magnesite is located at pH 3.4. In comparison, IEP values in the other slurries, with 0, 3, and 6 mass% of MgCO$_3$, are shifted below a pH value of 2. Below this pH value, the particle surface appeared to be positively charged, and the colloidal system becomes less stable [8] and less appropriate for the tape casting. The addition of dispersant led to further improvement of the zeta potential, as illustrated in Figure 4b in the case of NaHMP. Indeed, NaHMP allowed the adjustment of the zeta potential in the required range (absolute value of zeta potential >15 mV) when using a content ≥0.3 mass%.

*3.3. Rheological Properties*

3.3.1. Continuous Flow Tests

In general, the phyllosilicate suspensions stabilization depends on the concentration and charge density of dispersants as well as the specific interactions with clay platelets. The NaHMP and NaSi dispersants used in the present study are adsorbed onto the surface of KT2 particles, inducing electrostatic repulsive forces between the particles, which consequently affect the rheological behavior of the corresponding slurries. In order to determine the optimum dispersants content, the shear stress

and the viscosity of each slurry were investigated as a function of the shear rate for different dispersant contents (NaSi and NaHMP from 0.4 to 2 mass%; see Figure 5a,b). The flow curves were analyzed and fitted to the Herschel–Bulkley (H-B) model expressed by Equation (9), since it was the best model (compared to the Bingham and the Oswald de Waele models) in agreement with the associated high correlation coefficients $R^2$ (0.96 < $R^2$ < 0.99).

$$\tau = \tau_0 + K \times \dot{\gamma}^n \tag{9}$$

$\tau_0$ (Pa) the yield stress; $\dot{\gamma}$ ($s^{-1}$) the shear rate; K (Pa·s) the consistency; $n$, the power-law index (if $n < 1$, the fluid exhibits a shear-thinning behavior, and if $n > 1$, the fluid shows a shear-thickening behavior).

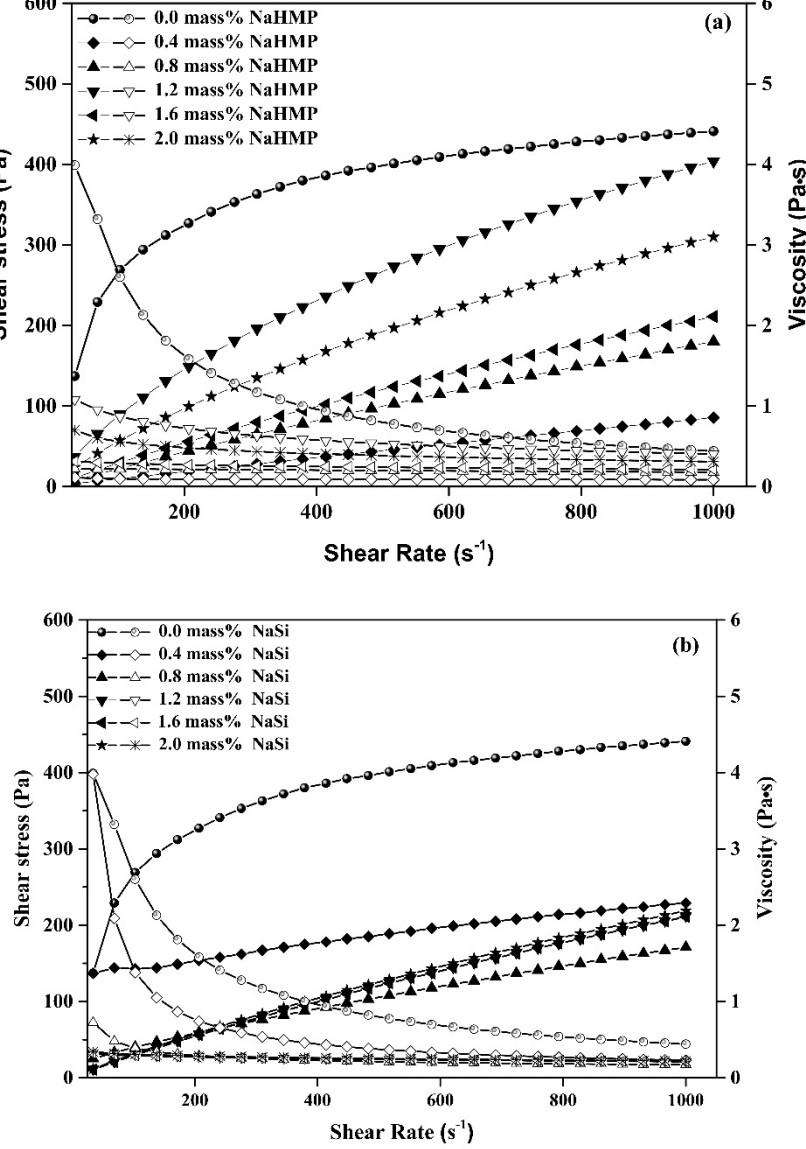

**Figure 5.** Flow curves (shear stress and apparent viscosity) of aqueous clay slurries with different amounts of (**a**) NaHMP (0.4–2 mass%) and (**b**) NaSi (0.4–2 mass%).

It can be observed from the evolution of consistency and yield stress shown in Figure 6, that the optimal dispersant content for NaHMP (Figure 6a) and NaSi (Figure 6b) are 0.4 and 1.2 mass%, respectively. However, the consistency (K) values decrease when adding the dispersants down to a minimum value, which corresponds to the optimal concentration of dispersants (0.4 and 1.2 mass%

optimal amount of NaHMP and SiNa, respectively). Further increases of dispersant above these optimized values leads to an increase of consistency, which is mainly due to the ions adsorption saturation of the clay particles.

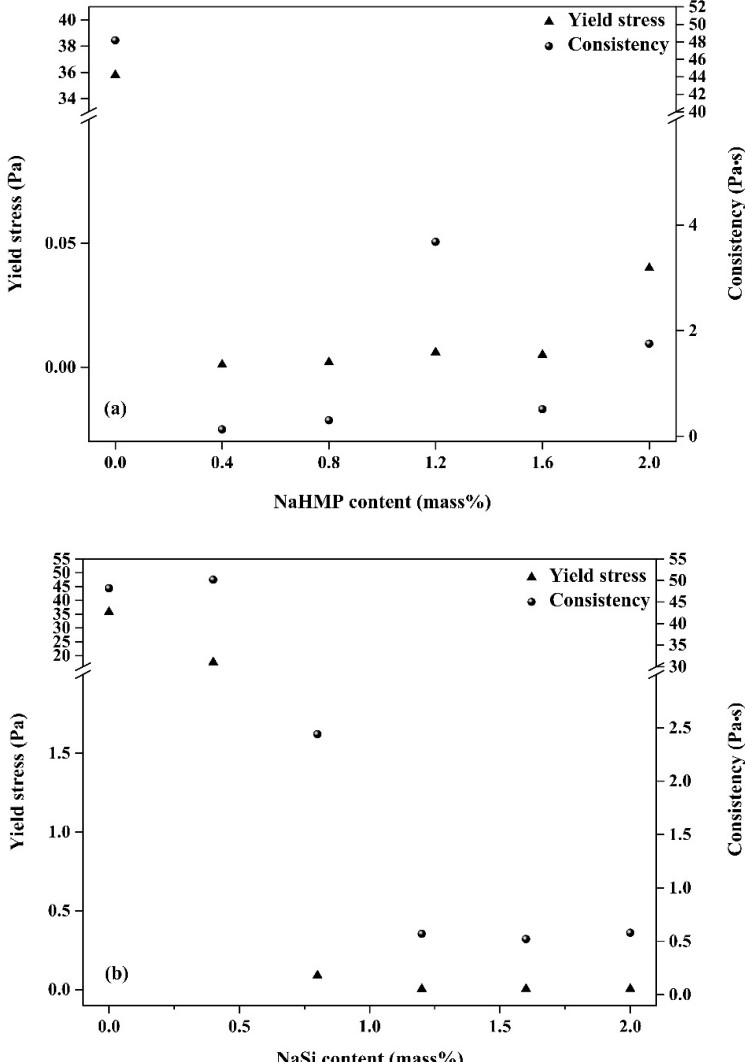

**Figure 6.** Yield stress and consistency of KT2 suspensions dispersed with (**a**) NaHMP and (**b**) NaSi at different concentrations (from 0.4 to 2 mass%).

For example, a significant increase in the yield stress occurs because of an increase in the amount of NaHMP dispersant from 0.4 to 2 mass%, corresponding to 0.001 and 0.040 Pa, respectively. It has been indicated that the dissolution of NaHMP in water involves the formation of $(PO_4)^{3-}$ and $(Na^+)$ ions, which are adsorbed onto the edge aluminol (Al–OH) sites [7] that give rise to a complex process between the dissolved phosphate $(PO_4)^{3-}$ and the alkaline earth cations [4]. This interaction mainly explains the yield stress increases of KT2 suspension dispersed using NaHMP. The pH values measured for all suspensions in the presence of NaHMP are greater than 4.5. According to Rao, F. et al. [38], the latter result indicates a good dispersion of KT2 particles.

In the case of the NaSi dispersant, below 1.2 mass% content, a higher yield stress and consistency were achieved. It was mainly due to the inter-particle bridging structures within the slurries. At ≥ 1.2 mass%, the KT2 suspensions display weaker yield stress and consistency values, which indicate a higher stability of the suspensions. Indeed, NaSi dispersant tented to inhibit edge-to-face attraction by electrostatic attraction, hydrogen bonds, Van der Waals attraction, and condensation [39]. In addition,

the power-law index of KT2 slurries was <1, which was characteristic of a shear-thinning behavior. Furthermore, the n values of the slurries with minimized consistency and yield stress seemed close to Newtonian behavior ($n = 1$), which is well correlated with the optimized suspensions using 0.4 mass% of NaHMP and 1.2 mass% of NaSi mass%. Moreover, it can be seen that the NaHMP dispersant is more efficient than NaSi due to its high molecular charge density.

According to the pH results, an increase in NaSi amount, the pH values of the suspensions increase from 5.45 to 8.86 for 0.4 and 2 mass% NaSi concentrations, respectively. This is mainly due to the interaction between edge and NaSi charges. Previous studies have indicated that the isoelectric point of kaolinite particle edges is in the range pH 3.9–5.6 [40,41]. At pH value ≤IEP (the edges of kaolinite particles are positively charged), edge-to-face and edge-to-edge associations should occur. Therefore, the optimized amount of NaSi corresponds to a pH value = 6.62 > IEP, with weaker yield stress. This result was also confirmed by zeta potential analysis.

The results of Landrou et al. [23] provided that MgO acts as an anti-dispersant for deflocculated kaolin slurries by stopping the activity of the NaHMP dispersant. However, many studies indicated that magnesite and magnesium hydroxide are usually calcined to produce MgO [42,43]. Thus, in this study, various $MgCO_3$ amounts were introduced into the mixed KT2/dispersants in order to study their effects on the rheological behavior of dispersed slurries with an optimal amount of NaSi or NaHMP dispersants. Regardless of the dispersant used, it was observed that the viscosity of dispersed slurries increases with increasing $MgCO_3$ content (3, 6, and 12 mass%). The results obtained with NaHMP are presented in Figure 7. The same trend was obtained with suspensions dispersed using the optimal amount of NaSi.

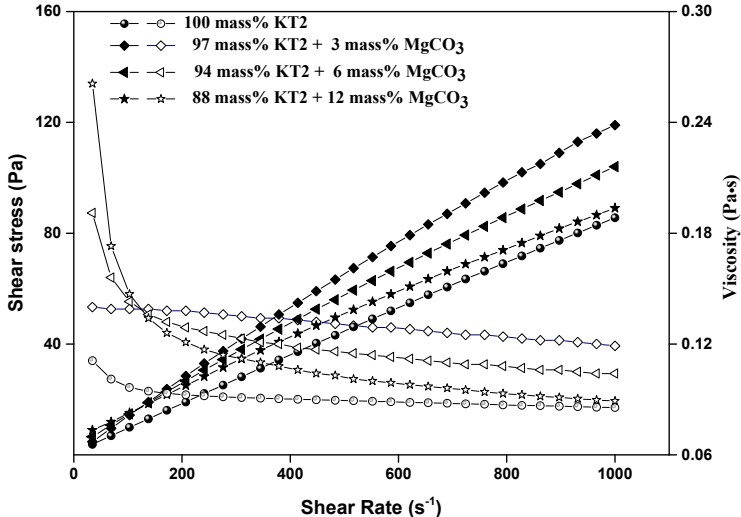

**Figure 7.** Flow curves and viscosity of KT2 + magnesite slurries with 0.4 mass% HMPNa (optimal concentration of dispersant).

From Figure 8, it can be seen that H-B yield stress and constancy values of dispersed suspensions by NaHMP or NaSi increased from 0.001 to 0.18 Pa and from 0.004 to 0.03 Pa, respectively, when increasing the $MgCO_3$ from 3 to 12 mass%. This can be explained by the inhibition of the electrostatic interaction between KT2 particles and NaHMP or NaSi dispersants because of the formation of soluble complexes between $Mg^{2+}$ cations and $(PO_4)^{3-}$ and $SiO_3^{2-}$ anions (Equations (10) and (11)).

$$3Mg^{2+} + (PO_4)^{3-} \rightarrow Mg_3(PO_3)_2 \tag{10}$$

$$Mg^{2+} + SiO_3^{2-} \rightarrow MgSiO_3 \tag{11}$$

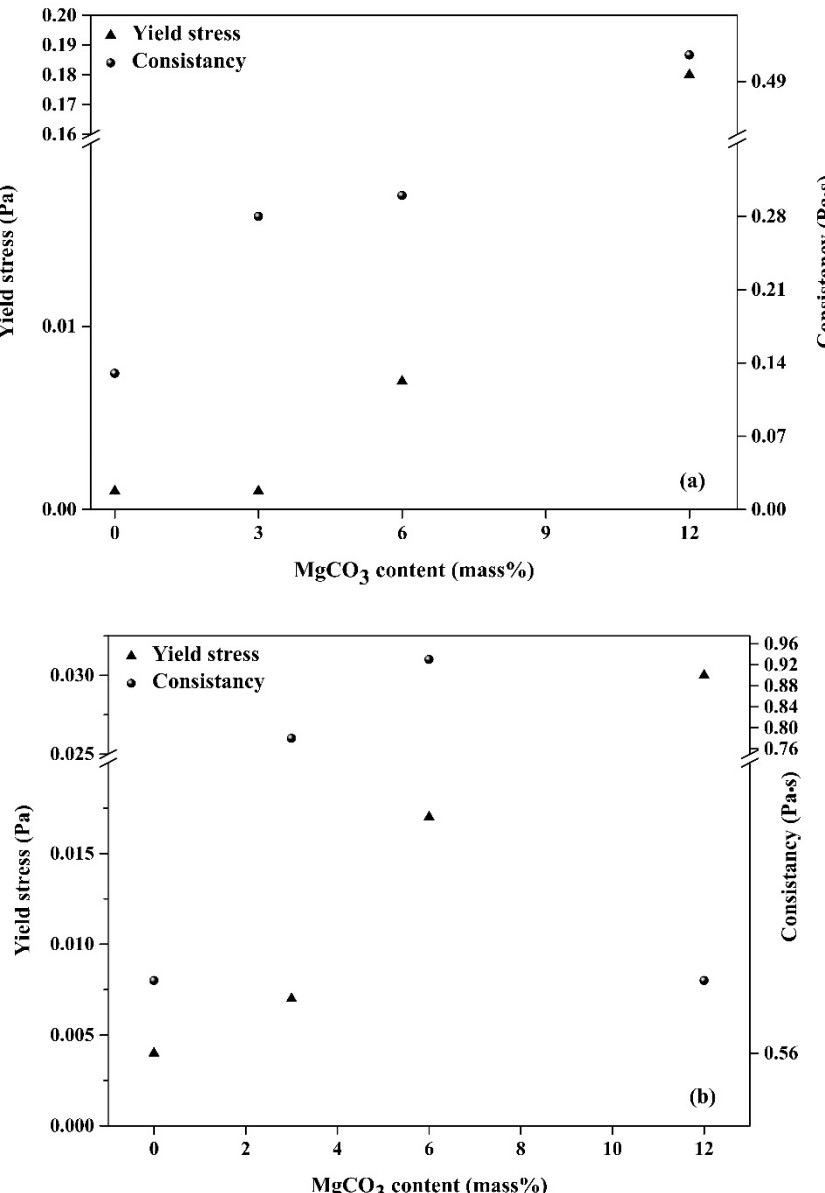

**Figure 8.** Yield stress and consistency of KT2/MgCO3 mixtures with (**a**) 0.4 mass% NaHMP and (**b**) 1.2 mass% NaSi.

However, the pH values of all KT2/MgCO$_3$ suspensions were around eight, which was mainly due to the higher value of the isoelectric point of MgCO$_3$ compared to the IEP of the kaolinitic clays (8 for MgCO$_3$, and 3.9 to 5.6 for kaolinitic clays) [44]. This difference in IEP and the significant solubility of MgCO$_3$ in aqueous media could generate a "hetero-coagulation" phenomenon (leading to coarse agglomerates formation), occurring through the interaction between particles of kaolinite and magnesite in slurries [45,46].

### 3.3.2. Oscillatory Rheological Tests

The oscillating shear experiments were performed in order to characterize the internal structure of the slurries and to drive more details about the viscoelastic behavior of kaolinitic clay and magnesite suspensions. The KT2/MgCO$_3$ suspensions were exposed to small oscillatory amplitude (i.e., clockwise then counter-clockwise) during the oscillation stress or strain sweep.

Therefore, in order to define the linear and non-linear viscoelasticity domains of the studied suspensions, the evolution of both the storage modulus ($G'$) and the loss modulus ($G''$) were investigated regarding the applied shear stress in the range 0.01 to 10 Pa at a constant frequency of 10 s$^{-1}$ (Figure 9).

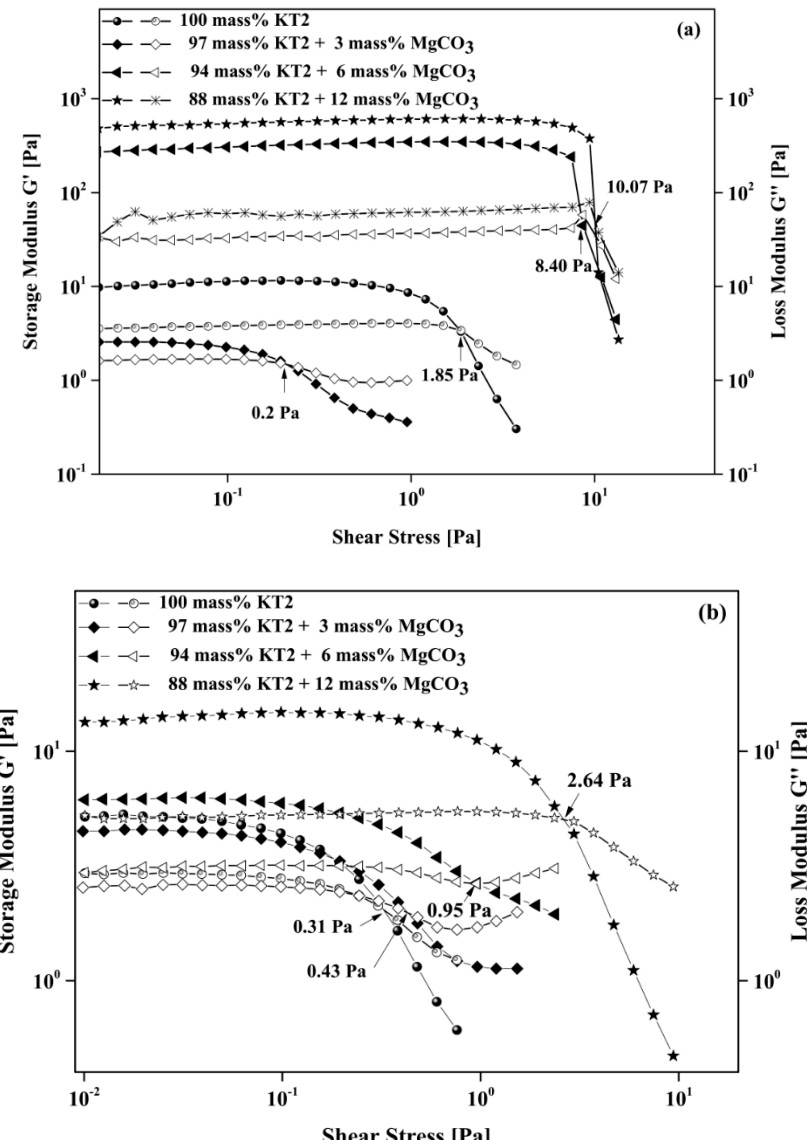

**Figure 9.** Stress sweep test at 10 s$^{-1}$ of KT2 and MgCO$_3$ mixtures dispersed with (**a**) 0.4 mass% of NaHMP and (**b**) 1.2 mass% of NaSi dispersants.

Consequently, the crossover point between $G'$ and $G''$ separates these two regions. Beyond this point, the rupture of the three-dimensional structure or the complex formed by the particles may occur. The results showed that at low oscillation stress, both moduli remained almost constant, and $G'$ was greater than $G''$ in the presence of each dispersant (NaHMP or NaSi). This result indicates that the suspensions behave in a more elastic than viscous manner since the small shear stress oscillation demanded minor structure deformation. The values of $G'$ and $G''$ were both constant up to a crossover point, and below this critical point, the system behaved in a linear viscoelastic manner (shear stress <0.1 Pa). When the shear stress increased above the crossover point, the viscous modulus dominated the elastic modulus and the system exhibited in a non-linear viscoelastic behavior (shear stress values >0.1 Pa). In this region, the storage modulus shows a drastic drop due to the breakdown of the weak attractive forces between the particles by the external shear stress, which destroys the internal

structure. In addition, the crossover point increased significantly when the amount of $MgCO_3 > 3$ mass%. Without $MgCO_3$, it is at 1.85 and 0.31 Pa for dispersed suspensions with an optimal amount of NaHMP and NaSi, respectively. However, at 6 and 12 mass% of added $MgCO_3$, the crossover point was equal to 8.4 and 10.06 for dispersed suspensions with NaHMP and 0.95 and 2.64 Pa for dispersed suspensions with NaSi. Figure 10 illustrates the differences in LVER with and without magnesite and the related particle arrangements within the suspensions.

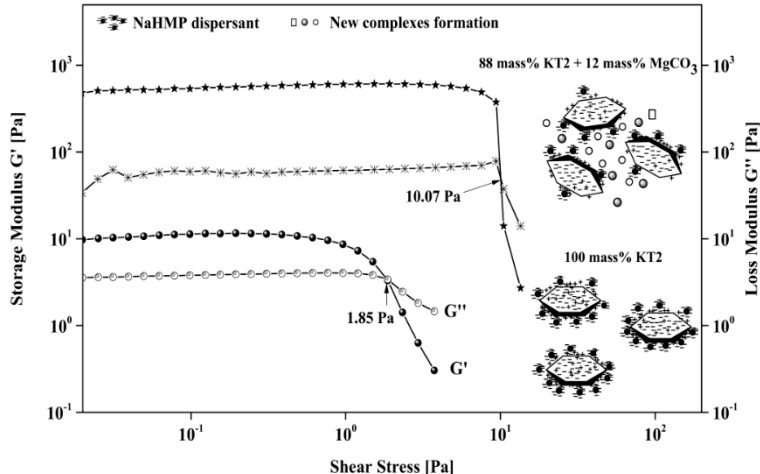

**Figure 10.** Schematic of the $MgCO_3$ influence on the $G'$ and $G''$ moduli for dispersed suspensions with and without magnesite (12 mass%).

The cohesive energy ($E_c$) is a parameter used to determine the stability of the internal structure of a multiphase system and the extent of the flocculated particles in a suspension. The higher values of cohesion energy involve the higher stability of the medium. From literature, several researchers have explained the construction used to determine the cohesion energy [47–49]. However, from the result of this study, it can be seen that $Ec$ strongly depends on the concentration of $MgCO_3$ and on the nature of the dispersants (Table 2). The increasing magnesite amount in the KT2 suspensions induces a significant decrease of the cohesive energy in the presence of NaHMP dispersant (from 1273.6 to 887.2 mJ/m$^3$ for 0 and 12 mass% of $MgCO_3$). This indicates that the $MgCO_3$ reduces the electrostatic interaction between KT2 particles and NaHMP, leading to the formation of soluble complexes (Figure 10). In the presence of NaSi dispersant, the $E_c$ is less sensitive to magnesite content up to 6 mass%, but it is highly increased when adding 12 mass% of magnesite ($Ec$ from 34.7 to 1364.7 mJ/m$^3$ for slurries with 0 to 12 mass% of $MgCO_3$, respectively). The enhancement of $Ec$ is mostly due to the increase of surface charge density and electrostatic repulsion between particles within the slurries. The results of this analysis confirm the previous assumptions about the formation complexes when adding magnesite amount in the KT2 suspensions.

The frequency sweep tests for various KT2/$MgCO_3$ mixture dispersions consist of measuring the evolution of both moduli as a function of frequency in the range 0.1 to 100 s$^{-1}$ and fixing the stress at values belonging to the linear viscoelasticity domain (0.1 Pa). The frequency sweeps allowed determination and investigation of the structuring mechanisms present in a fluid at relaxation time. Therefore, in this domain, the rheological properties are not strain- or stress-dependents. The frequency sweep analysis of slurries was plotted (Figure 11) without and with magnesite (0, 3, 6, and 12 mass%), which were dispersed with an optimal amount of dispersant (0.4 mass% NaHMP or 1.2 mass% NaSi).

**Table 2.** Cohesive energy parameters for all suspensions mixtures (KT2/MgCO$_3$).

| Dispersants | Suspensions with Magnesite Addition (mass%) | Critical Modulus $G'c$ (Pa) | Critical Deformation (%) | Cohesive Energy $Ec$ (mJ/m$^3$) |
|---|---|---|---|---|
| NaHMP | 0 | 14.4 | 13.3 | 1273.6 |
| | 3 | 2.7 | 3.5 | 16.5 |
| | 6 | 339.4 | 2.3 | 897.7 |
| | 12 | 606.8 | 1.7 | 887.2 |
| NaSi | 0 | 4.8 | 3.8 | 34.7 |
| | 3 | 4.5 | 3.5 | 27.6 |
| | 6 | 6.1 | 3.1 | 29.3 |
| | 12 | 15.2 | 13.4 | 1364.7 |

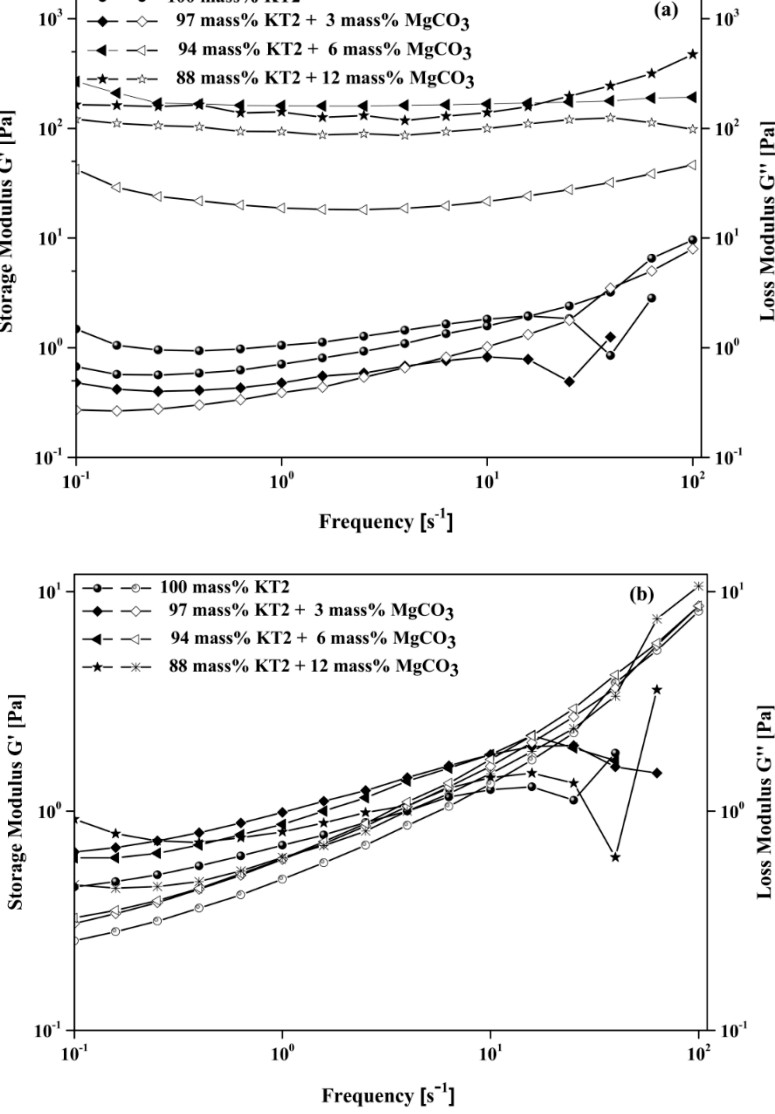

**Figure 11.** Frequency sweep test at 0.1 Pa shear stress of KT2 and MgCO$_3$ mixture dispersed with (**a**) 0.4 mass% NaHMP and (**b**) 1.2 mass% NaSi dispersants.

As expected, in the regions of low frequency, $G'$ always remains higher than $G''$ for all mixture slurries and slightly increased with increasing the applied frequency in the presence of either dispersant.

In contrast, their evolution mostly depended on the nature of dispersant and amount of $MgCO_3$. This means that when the amount of $MgCO_3$ was close to a critical value (6 mass%) and/or when the molecular weight of dispersant increased (NaHMP > NaSi), the rheological behavior of the aqueous dispersion became highly elastic and plastic. In addition, the results show that beyond a certain frequency, the $G''$ becomes greater than $G'$, indicating that the viscous contribution starts to become pre-significant above a threshold frequency.

According to the rheological results, the $KT2/MgCO_3$ suspensions dispersed with an optimal amount of NaHMP (0.2 mass%) and with 5 mass% of binder and plasticizer presented (at specific shear-rate of 10 s$^{-1}$) an apparent viscosity between 0.3 and 1 Pa·s, which is required for the tape casting process (the ratio binder/plasticizer being equal to 1 by weight). Indeed, consistencies obtained at a shear rate of 10 s$^{-1}$ were 0.37, 0.45, 0.49, and 0.73 Pa·s for 0, 3, 6, and 12 mass% added $MgCO_3$, respectively. As expected, before casting, all suspensions with and without $MgCO_3$ exhibited shear-thinning behavior. Figure 12 illustrates the typical tapes obtained for KT2-magnesite mixtures before and after drying at room temperature. These tapes appear free of cracks and any critical defects, and therefore, the optimized slurries are appropriate for the tape casting process.

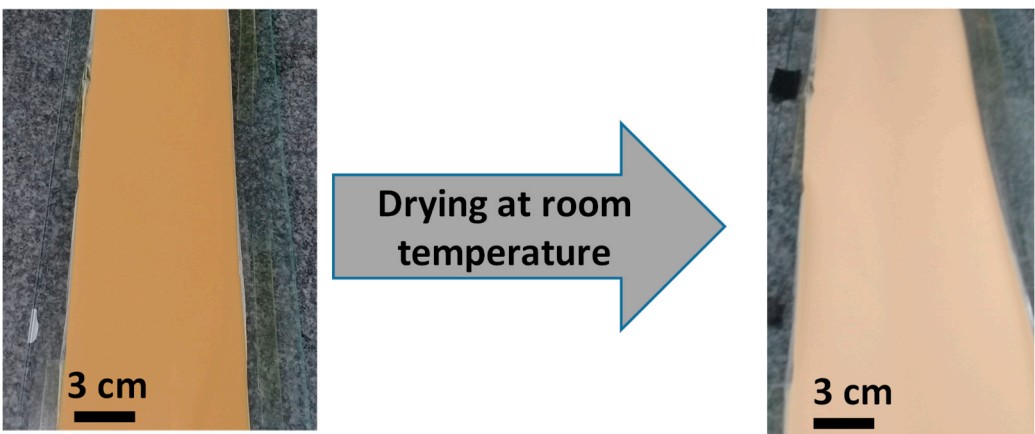

**Figure 12.** Photograph of $KT2/MgCO_3$ green tapes (6 mass% of $MgCO_3$) before and after drying at room temperature.

## 4. Conclusions

The purpose of the present study was to investigate the behavior of suspensions of an Algerian kaolinitic clay (KT2) with the addition of magnesite ranging from 0 to 12 mass%. Indeed, the physical and chemical characteristics were determined, and the stability of these suspensions was studied through rheological (flow and oscillatory modes) measurements.

The chemical and XRD analyses of the raw materials (kaolinite clay and magnesite) indicated the presence of (i) three crystalline phases in the KT2 sample, namely, kaolinite, muscovite, and quartz; (ii) magnesite as the major phase in the magnesite, associated with secondary phases like quartz and dolomite.

The DTA and TG curves described the different thermal transformation characteristics of the raw materials and the mixture samples. As expected, the main transformations in the range 400 °C to 700 °C were the dehydroxylation of kaolinite into metakaolinite and the decomposition of $MgCO_3$ to the MgO and $CO_2$, respectively. Within the samples containing 6 and 12 mass% of $MgCO_3$, the maximum of the endothermic peak position previously located at 480 was shifted to 518 °C.

The measurements of the zeta potential appeared helpful in achieving the correct pH conditions for the various slurries regarding the improvement of their dispersion and stability. Indeed, the addition of magnesite induced an increase of the zeta potential of the studied KT2 slurries; nevertheless, an absolute value of zeta potential >15 mV was more appropriate for the processing of tape casting slurries.

Under continuous shear, all investigated slurries indicated a shear-thinning behavior. The best stabilization results of KT2 dispersions can be obtained while using optimal amounts of NaSi or NaHMP, 1.2 and 0.4 mass%, respectively. This is related to the high electrosteric repulsions between the particle surfaces. Without dispersant, the yield stress and consistency of suspensions were very important (35.3 Pa), which confirmed that the E–F structure (coagulated) dominates the system. In the presence of either dispersant, the H-B yield stress and consistency values increased with the $MgCO_3$ increasing, which is mainly due to the inhibition of the electrostatic interaction between KT2 particles and NaHMP or NaSi dispersants.

The examination of the viscoelastic properties of the studied suspensions allowed determining the limit of the linear viscoelasticity region, which was found under 0.1 Pa for all dispersions. At low amplitude stress, $G'$ was greater than $G''$, and both moduli remained almost constant. The dispersions behave in a more elastic than viscous manner. Increasing the amplitude stress over the crossover point causes the viscous modulus to dominate the elastic modulus, and the system behaves in a non-linear viscoelastic manner (shear stress values >0.1 Pa).

Frequency sweep tests showed that at low frequencies, the elastic part was greater than the viscous one, i.e., $G' > G''$, and on the other hand, $G'' > G'$, at high frequencies. The addition of $MgCO_3$ above the critical concentration limit of 6 mass% gave rise to the formation of a stronger gel phase and the aqueous dispersion becoming highly elastic and plastic. Moreover, it can be seen that increasing the magnesite amount in the KT2 suspensions induces a significant decrease of the cohesive energy, which was attributed to weak electrostatic interactions between the solid particles.

**Author Contributions:** For the present research paper, all the co-authors have contributed in different steps from the experimental work, the administrative and scientific support, the discussion, to the writing and supervision. More precisely the contributions can be summarized as follows: "conceptualization, F.Z. and G.L.-N.; methodology, F.Z. and G.L.-N.; software, A.H., N.A., I.D.; validation, G.L.-N., and F.Z..; formal analysis, A.H., N.A. and G.L.-N.; investigation, A.H., I.D., N.A..; resources, C.P., G.L.-N.and F.Z.; data curation, A.H., I.D., N.A. and G.L.-N.; writing—original draft preparation, A.H., N.A. and G.L.-N.; writing—review and editing, A.H. and G.L.-N.; visualization, G.L.-N.; supervision, F.Z. and G.L.-N.; project administration, F.Z., G.L.-N. and C.P.; funding acquisition, F.Z."

**Funding:** This research received no external funding.

**Acknowledgments:** The authors gratefully acknowledge the Algerian kaolin company "Soalka", Tamazert, Jijel and faïencerie of Thenia, Boumerdes (3500), Algeria for their help in supplying the kaolin and magnesite, respectively.

**Conflicts of Interest:** The authors declare no conflict of interest.

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
