# Peer review of "Kaolinite-Magnesite Based Ceramics. Part I: Surface Charge and Rheological Properties Optimization of the Suspensions for the Processing of Cordierite-Mullite Tapes"

_minerals, doi:10.3390/min9120757_

Round 1

Reviewer 1 Report

see file attached.

Author Response

Responses/Comments to reviewers

Paper reference: minerals-640381

Title: Kaolinite-magnesite based ceramics. Part I: Surface Charge and Rheological Properties Optimization of the Suspensions for the Processing of Cordierite-mullite Tapes

Dear Editors and Reviewers,

Thank you for your different comments and remarks on the manuscript that we have submitted to the journal “Minerals”.

Kindly find below our answers and comments to the reviewers reports.

Sincerely,                                                                                                                          Gisèle Lecomte-Nana et al.

Reviewer 1

Submission Date

28 October 2019

Date of this review

08 Nov 2019 14:08:22

A basic assumption in the manuscript on kaolinite-magnesite based ceramics is that surface charge of particles involved in this system is important. This begins in the Abstract with the descriptions of pH-dependencies of zeta potential and effects on microstructure. In the text it is highlighted and measured that kaolinite has a low iep and magnesite a higher iep, which can induce heteroaggregation at specific mixing ratios. Surface charge of mineral compounds is systematically shifted by the addition of two different polyanions. Based on this strong focus on charge properties in the manuscript and due to the fact that considering the role of zeta potential is somehow unique in these kind of manuscripts my suggestion is that the “adjustment of surface charge properties”, performed in this study, is highlighted in the title of the manuscript already. This will make the nice content much better visible. My suggestion is to publish this manuscript in `minerals´ after revision.

Response: We have modified the title as “Kaolinite-magnesite based ceramics. Part I: Surface Charge and Rheological Properties Optimization of the Suspensions for the Processing of Cordierite-mullite Tapes”

Line 18: for clarity better to write “kaolinitic clay”

Response: Correction done

Line 41: in petrography you have primary minerals from magmatic and metamorphic origin and

secondary minerals, oxides and clay minerals, formed by alteration and weathering. The term

secondary minerals is not applicable here.

Response: Correction done

Line 43: remove “maybe”

Response: Correction done

Line 48: better to write “discrete particles”

Response: Correction done

Line 52: when a negatively charged dispersant

Response: Correction done

Line 53: it adsorbs on the the external particle surface and makes the surface charge more negative.

Response: Correction done

Line 55: Na2SiO3

Response: Correction done

Line 56-58: Here the mechanisms for anion adsorption on mineral surfaces are not fully described.

These anions can adsorb to the central cations Si and Al and increase the coordination number

hereby. As multivalent anions free negatively charged sites shift zeta potential to more negative

values.

Response: We have included the sentence  “On the edges, these anions can adsorb to the central cations (Si and/or Al) and increase the coordination number hereby. As multivalent anions, free negatively charged sites shift zeta potential to more negative values”.

Line 64: What do you mean with “particle interactions”, differences in zeta potential?

Response: We mean the interactions between the particles within the slurries, mainly clay particles. We have added in the text “(edge-edge, face-face or edge-face interactions)”

Line 77: Here you discuss MgO. In your study you use MgCO3. Here I do not know what can be

learnt here from MgO as it is not stable in aqueous suspension due to the reation to Mg(OH)2 and

subsequently MgCO3.

Response: We have indicated in these general considerations in order to have a general understanding of the MgO based systems according to the existing literature. Thereafter, we have focus on the case of magnesite, and the detailed reactions in aqueous media have been also in paragraph 3.2, reactions R5 to R7.

Line 111: What is the energy applied by ultrasonication?

Response: Correction done

Line 117: Bruker

Response: Correction done

Line 135: You apply very extensive milling for 16 h. What is the effect of milling on particle size

and dislocations. Some orientation is welcome.

Response: Correction done

Line 197: BET surface was estimated with 11.68 m²g-1. What is the error of the method. Why it is

an estimation and not a quantification.

Response: Correction done. We have added the error related to the measurement of the BET SSA

Table 1: replace BET by SSA (specific surface area).

Response: Correction done

From particle size distribution shown in Figure 3 the magnesite occurs much finer than kaolinite. In Table 1 the SSA is smaller for the finer sized magnesite and larger for the coarse sized kaolinite. It should be the other way round. Is there an explanation based on particle morphology? Before sizing, did you add a dispersant and performed sonication for improving dispersion?

Response: Clay minerals are characterized by different ranges of SSA and the value obtained here is in agreement of expected behavior for kaolinitic clays (from 15 to 30 m².g-1). Also for the magnesite, the obtained value is the mean value upon three measurements. Also note that all particle size distribution analyses were performed after dispersion using dispersant and sonication as stated in paragraph 2.2. We have also modified the comment of figure 3 and included SEM images of raw materials in paragraph 3.1: “The latter trend is correlated with SEM observations of KT2 and magnesite as shown on Fig 3 (a) and (b). Note that the platelets and tubular-like shape of the kaolinitic clay particles involve the stacking of several clay mineral sheets. This fact could justify the higher SSA obtained with KT2 compared to the SSA value of magnesite.”

Figure 4: Here it would be nice to have the curve not only for kaolinite and the mixtures with

magnesite, but also for the other end-member pure magnesite. This will much better show the effect of the magnesite admixture.

Response: The target of the study was to investigate the effect of magnesite addition on the behavior of the kaolinitic clay. Indeed, the measurements performed on kaolinitic clay slurries containing 0, 3, 6 and 12 mass% of magnesite had shown no linear zeta potential variation regarding the magnesite content. We think that the measurement of the zeta potential of magnesite may be useful for further studies if the magnesite amount is higher than 25 mass%.

Line 230: Unstable by an increase of repulsive forces?

Response: Correction done. We have indicated less stable according to the slurry.

Line 286: Here you start again with MgO. More clarification is needed as this is not stable in

aqueous suspension.

Response: As stated in the response of Line 77, we considered MgO based system and then focus on the case of magnesite. Indeed, the reactions showed the species that are stable in aqueous suspensions. Thus some similarities.

Line 310: There is more recent literature available on heteroaggregation considering also size

effects: Dultz et al. 2019 Size and charge constraints in microaggregation: Model experiments with mineral particle size fractions. Applied Clay Science 170, 29-40.

Response: Correction done. The reference has been included.

Conclusions: The conclusions represent in their present form an extended summary of the

manuscript. This content should be added in the Abstract. A topic for the conclusion is for example to discuss if everything was done here for the optimization of the system, if future work is needed here and if it is useful to consider zeta potential in such kind of studies.

Response: Correction done. The conclusion and abstract have been modified by including the following: “The zeta potential of the kaolinitic clay slurry matched the requirements for tape casting. The addition of magnesite in the kaolinitic slurries tended to increase the zeta potential towards the required limit values. Despite, the further addition of surfactants allowed improving the zeta potential in agreement with the tape casting conditions.” in the abstract; “The measurements of the zeta potential appeared helpful in achieving the correct pH conditions for the various slurries regarding the improvement of their dispersion and stability. Indeed, the addition of magnesite induced an increase of the zeta potential of the studied KT2 slurries; Nevertheless an absolute value of zeta potential > 15 mV was more appropriate for the processing of tape casting slurries.” In the conclusion.

Reviewer 2 Report

In this work, authors report their work Kaolinite-magnesite based ceramics. Part I: The Optimization of the Formulation for the Processing of Cordierite-mullite Tapes. Before this manuscript is published, I have some comments and concerns which authors should respond to.

What is the influence of sodium silicate concentration on the morphology. The surface area was higher or lower compared to the original clays. How was the morphology of these two solid catalysts observed by SEM or AFM?

Do the authors have microscopic with EDX mapping the solid catalysts after sodium silicate formulation of cordierite mullite.

Please correct on the Table 1Raw material line are given on assignments of your Table.

Complete characterization of the solids obtained after dried admixtures must be given to accomplish the real effect of the dispersant agents against kaolinite-magnesite surfaces.

Author Response

Responses/Comments to reviewers

Paper reference: minerals-640381

Title: Kaolinite-magnesite based ceramics. Part I: Surface Charge and Rheological Properties Optimization of the Suspensions for the Processing of Cordierite-mullite Tapes

Dear Editors and Reviewers,

Thank you for your different comments and remarks on the manuscript that we have submitted to the journal “Minerals”.

Kindly find below our answers and comments to the reviewers reports.

Sincerely,                                                                                                                          Gisèle Lecomte-Nana et al.

Reviewer 2

Submission Date

28 October 2019

Date of this review

15 Nov 2019 18:17:30

In this work, authors report their work Kaolinite-magnesite based ceramics. Part I: The Optimization of the Formulation for the Processing of Cordierite-mullite Tapes. Before this manuscript is published, I have some comments and concerns which authors should respond to.

What is the influence of sodium silicate concentration on the morphology. The surface area was higher or lower compared to the original clays. How was the morphology of these two solid catalysts observed by SEM or AFM?

 Do the authors have microscopic with EDX mapping the solid catalysts after sodium silicate formulation of cordierite mullite.

Response: SEM and EDS were performed on the final sintered tapes containing cordierite and or mullite. Not on the slurries. Nevertheless, we have included the SEM images of the raw materials as figure 3 (a) and (b).

Please correct on the Table 1Raw material line are given on assignments of your Table.

Response: Correction done.

Complete characterization of the solids obtained after dried admixtures must be given to accomplish the real effect of the dispersant agents against kaolinite-magnesite surfaces.

Response: The first part was concerned with the slurries and we aimed at introducing the characteristics of the dried tapes in the second part of our paper whereby, we will compare the properties of green tapes and fired tapes. Also note that when the scope of the present paper was to optimize the slurry characteristics using rheology and zeta potential measurements prior to the tape casting. Therefore, only the optimized formulations were cast.